# The Role of Big Data in Promoting Green Development: Based on the Quasi-Natural Experiment of the Big Data Experimental Zone

**DOI:** 10.3390/ijerph20054097

**Published:** 2023-02-24

**Authors:** Jiangying Wei, Xiuwu Zhang

**Affiliations:** School of Statistics, Institute of Quantitative Economics, Huaqiao University, Xiamen 361021, China

**Keywords:** big data analytics, green total factor productivity, difference-in-differences with multiple time periods, quasi-natural experiment of the Big Data Experimental Zone, resource allocation

## Abstract

Against the backdrop of the pressing issue of global warming, the concept of green development, which emphasizes the rational utilization of resources and energy, has emerged as a viable model for future economic growth. Despite this, the interplay between big data technology and green development has yet to receive due consideration. This study aims to shed light on the role of big data in green development from the perspective of factor configuration distortion. To this end, a panel data analysis of 284 prefecture-level cities spanning from 2007 to 2020 was conducted, utilizing the Difference-in-Differences (DID) and Propensity Score Matching-Difference-in-Differences (PSM-DID) models to assess the impact of the establishment of the National Big Data Comprehensive Experimental Zone on green total factor productivity. The findings reveal that the establishment of the National Big Data Comprehensive Experimental Zone has a positive impact on green total factor productivity, primarily through optimizing the capital and labor allocation distortions, with the effect being more pronounced in areas with high levels of human capital, financial development, and economic activity. This research provides empirical evidence to evaluate the impact of the establishment of the National Big Data Comprehensive Experimental Zone and offers valuable policy implications for the pursuit of high-quality economic development.

## 1. Introduction

In recent years, a new generation of information technology with the Internet, big data, and cloud computing as its core has developed rapidly, accelerating the digital transformation of the economy and society, which is profoundly affecting the development direction of the world economy. In order to speed up the construction of a factor market, establish and improve the market-oriented mechanism of factors, and ensure the orderly circulation and efficient use of data factors under the protection of relevant systems, China issued the Action Plan for Promoting the Development of Big Data in 2015, proposing the construction of a national comprehensive experimental area for big data, and making the overall deployment of big data as an important national strategy. The National Big Data Comprehensive Experimental Zone mainly promotes the development of the regional big data industry by undertaking seven major tasks, including data resource management and sharing development, data center integration, data resource application, etc., and plays a leading role in the overall construction of the China data factor market.

How to give full play to the role of the National Big Data Comprehensive Experimental Zone in promoting high-quality economic development is a topic that scholars have studied deeply in recent years. The existing literature focuses on how the digital economy affects the total factor productivity of cities and firms [1,2], innovation ability [3], industrial structure upgrading [4], economic development mode, and other issues. However, there is still a lack of empirical studies to accurately evaluate the role of national big data comprehensive experimental zones in high-quality economic development. At present, there is no unified understanding and conclusion on how to help regional green total factor productivity in the National Big Data Comprehensive Experimental Zone. In view of this, this paper will focus on the following questions: Can the National Big Data Comprehensive Experimental Zone improve green total factor productivity? What is the mechanism behind it? What is the difference between cities in the effect of the National Big Data Comprehensive Experiment Zone on regional green total factor productivity?

The influence of the National Big Data Comprehensive Experimental Zone on green total factor productivity is multidimensional and compound. At the micro level, the deep application of digital technology helps to reduce the search cost, marginal cost, and matching cost of enterprises [5], improve the innovation capability of enterprises and promote the initiative of enterprises [6]. At the meso level, the establishment of the National Big Data Comprehensive Experimental Zone helps to accelerate the iteration of a new generation of digital and intelligent technologies and constitutes a benign interaction mechanism with the high-end development of the intelligent industry. At the macro level, the iterative application of the new generation of digital and intelligent technologies helps to optimize the policy environment and institutional system, stimulate the vitality of social innovation [7], improve the efficiency of resource allocation, and then improve the green total factor productivity.

To verify whether the National Big Data Comprehensive Experimental Zone has a driving effect on green total factor productivity, and clarify the laws and differences of this effect in time and different types of cities, this paper takes the National Big Data Comprehensive Experimental Zone as a quasi-natural experiment, builds the green total factor productivity index based on the SBM-GML index model, and studies the impact of the establishment of the National Big Data Comprehensive Experimental Zone on green total factor productivity by using DID with multiple time periods and the PSM-DID model. It is found that the establishment of the National Big Data Comprehensive Experimental Zone can effectively improve the green total factor productivity, and this effect has significant heterogeneity under different urban characteristics. At the same time, the establishment of the National Big Data Comprehensive Experimental Zone and the distortion of factor allocation have significant interactions with green total factor productivity.

The possible marginal contributions of this paper lie in: Firstly, the SBM-GML index model is used to construct green total factor productivity, considering the negative impact of unexpected output, focusing on the analysis of the role of big data in the green economic development, and providing a reference for China’s green economic transformation policy from the perspective of digital economic development. Secondly, based on the realistic perspective of factor allocation distortion, the labor factor and capital factor distortion index are introduced to study the impact and mechanism of the establishment of the National Big Data Comprehensive Experimental Zone on green total factor productivity. Finally, it is further confirmed that the impact effect of the establishment of the National Big Data Comprehensive Experimental Zone is heterogeneous in terms of urban characteristics, which provides a reference for exploring the realization path of high-quality development of the regional economy.

## 2. Literature Review and Research Hypothesis

### 2.1. “Big Data” Concept and Policy Evolution

In the 1990s, “big data” began to appear. At this time, the definition of “big data” was when the amount of data was too large for memory, local disks, or even remote disks to handle. When Google discovered and applied the definition of cloud computing in 2006, the value of “big data” really appeared. In 2012, China released the “Twelfth Five-Year National Government Information Construction Project Plan” involving types of data from databases relating to population, law, space, macroeconomics, and culture, and began to enter the era of open, shared, and intelligent big data. In 2013, big data began to be used in various industries, which set off social development and changes and had an important impact on the existing relationship between productivity and production. With the development of digital technology, the application scenarios of big data have increased, and the concept of “big data” has gradually developed from a large amount of data to a comprehensive concept including data amount, technology, and application. The definition of big data in the Action Plan for Promoting the Development of Big Data issued by the State Council in 2015 holds that big data is data with the characteristics of large data volume, diversified structure, and strong timeliness, in which value is the core of big data, and data, technology, and application as three elements of big data revolve around value (Zhu and Xiong, 2015) [8].

With the rapid development of information technology represented by big data, countries have gradually attached importance to the development of the big data industry, and developed economies such as the United States, Japan, and the European Union have formulated a series of strategic action plans related to big data. In 2019, the US Office of Management and Budget and the Ministry of Commerce jointly issued the “Federal Data Strategy and 2020 Action Plan” to develop data as a strategic resource, and the focus on data shifted from technology to capital. In 2012, the US government issued the Big Data Research and Development Plan, and established the “Big Data Senior Steering Group”. In 2019, the US Office of Management and Budget and the Ministry of Commerce jointly issued the “Federal Data Strategy and 2020 Action Plan” to develop data as a strategic resource, and the focus on data shifted from technology to capital. In 2012, Japan launched the Comprehensive ICT Strategy for 2020, and put forward the goal of “Japan active in the ICT field”, focusing on the application of big data. In 2013, it announced the Declaration on Creating the Most Cutting-edge Information Technology Country, aiming to build Japan into a “world-class society that widely uses information industry technology”. In 2015, China raised big data to the level of a national strategy and adopted the Action Outline on Promoting Big Data Development. In 2016, the state approved the establishment of a National Big Data Comprehensive Experimental Zone in Guizhou and promoted the development of big data by taking the whole province of Guizhou as an “experimental field” to explore the reference experience that can be used to implement the national big data strategy.

### 2.2. Green Economic Development

With the continuous and rapid development of the national economy, environmental pollution, ecological damage, and resource waste have followed, and the World Air Quality Report (2018) shows that some Chinese cities have been included in the key pollution monitoring cities. Therefore, a development path that balances productivity and green sustainability has become an important issue of concern for all sectors of society. Green total factor productivity (TFP) is a concept to bring environmental and resource factors into the productivity analysis framework, which is suitable for explaining the mystery of China’s green economic growth (Li et al., 2015) [9]. The promotion of green TFP will help China’s high-quality economic development and meet the requirements of sustainable economic and environmental development. Therefore, the main influencing factors affecting green economic growth need to be further clarified (Fan et al., 2015) [10]. Feng et al. (2021) [11] used a difference-in-difference (DID) model to find that China’s carbon emissions trading system directly contributes to green total factor productivity at the national level, with significant spatial heterogeneity at the regional and local levels. Zhang et al. (2021) [12] used a spatial double difference model to carry out an empirical test and found that environmental decentralization in China increased urban green patent grants, but did not significantly increase urban green productivity. Zhang and Li (2020) [13] report that there is a reverse green technology spillover effect in China’s foreign direct investment. 

However, no matter the research on energy saving and emission reduction policies, fiscal decentralization system, or opening to the outside world, the widespread existence of administrative monopoly factors in the factor market in the process of China’s market economic system reform has been neglected, which is an important factor causing the distortion of the factor market. Although the distortion of the factor market can improve the macro-control function of the government in the short term, it will not only inhibit economic growth but also lead to more serious environmental pollution problems in the long run. Therefore, the distortion of the factor market is an important factor that cannot be ignored in analyzing the growth of the green economy. Specifically, according to the research of Zhang et al. (2020) [14] and Bai and Bian (2016) [15], it is found that external government intervention under the fiscal decentralization system and the “promotion tournament” mode will reduce the production cost of enterprises by depressing the prices of tangible factors such as labor, and, on the one hand, attract capital and enterprises to enter, but on the other hand, destroy the formation mechanism of production factor prices, trigger the distortion of factor markets, lead to the development of highly polluting industries, and then inhibit the growth of the green economy. In the case of a distorted factor market, local governments artificially suppressing the prices of factors such as capital will lead enterprises to be more inclined to produce tangible factors and inhibit the enthusiasm of independent innovation of enterprises (Chen and Jiang, 2020) [16]. In addition, the distortion of the factor market will lead to excessive use and waste of energy and hinder the growth of the green economy. The empirical study by Lin and Du (2013) [17] found that if distortion of the factor market is eliminated, the energy efficiency of China can be improved by 10%.

In summary, it can be seen that, on the one hand, the existing literature is less interested in incorporating economic growth and environmental pollution into the same analytical framework at the same time, and the research on green economic growth focuses on technological progress, opening up to the outside world, and R&D investment, etc. Few literature studies discuss green economic growth from the perspective of big data. On the other hand, few scholars have included factor market distortions in the analytical framework of big data affecting green economic development. This paper examines the issue of green economic development from the perspective of big data and creatively introduces factor market distortion into the analysis of green economic development in an attempt to analyze the relationship between big data development, factor market distortion, and green total factor productivity, and to “add bricks and mortar” to the high-quality economic development.

### 2.3. Big Data and Green Economic Development

Green economic development, with environmental sustainability and economic development at its core, not only reflects the continuous improvement of production efficiency but also emphasizes a production method that benefits the natural environment. Different from the traditional production technologies employed in the past, digital technology characterized by big data has the characteristics of high technology content and low environmental cost and pays more attention to coordinated development with the environment while improving production efficiency (Arik, 2009) [18]. Regarding the measurement of big data, some scholars measure it by conducting questionnaires on related enterprises, but this measurement is easily influenced by the differences in survey sample selection and subjective opinions of survey enterprises. Subsequently, quantifying the level of big data with statistical data attracted attention. For example, Fang et al. (2021) [19] used principal component analysis to measure the development level of big data in various provinces and cities. However, the method measured from multiple perspectives such as Internet penetration rate and ICT, which did not reflect the characteristics of big data itself. Models with the double difference method can avoid the endogeneity problem to a greater extent and focus the research question on the specific impact of big data policy itself, using the implementation of policies such as the “National Big Data Comprehensive Experimental Zone” as a natural experiment. At present, there are relatively few studies on the impact of big data on the development of a green economy, and the related studies mainly focus on organizational performance (El-Kassar and Singh, 2019) [20], green supply chain management (Chen and Liu, 2020) [21], and energy efficiency and energy management (Wu et al., 2016) [22]. There are great differences in the related literature on the role of big data in the development of a green economy. One point of view holds that the promotion of big data to green development is not significant or even has a certain inhibitory effect. For example, Ghasemaghaei and Calic (2020) [23] found that the increase in the amount of big data can not effectively improve the innovation efficiency of enterprises. Another view is that big data can significantly improve the development of a green economy. Xu et al. (2019) [24] believe that the rapid development of big data provides an important way for the green development of China. The remarkable advantages of the National Big Data Comprehensive Experimental Zone in technological innovation and application, data resource management and sharing, data center integration, and data resource application can effectively deal with the contradiction between the supply of and demand for factors and the lag in management efficiency in high-quality economic development.

Big data empowers green total factor productivity in three main ways: First, the digital service brought by the establishment of the National Big Data Comprehensive Experimental Zone can promote the emergence of new industries, new formats, and new models [25], and help enterprises in the initial stage and growth stage in this region break through the production bottleneck, effectively solving the problems of rising employment costs and decreasing profit margins of manufacturing enterprises. Second, the information platform sharing brought about by the establishment of the National Big Data Comprehensive Experimental Zone can effectively reduce the cost of searching, trading, matching, and copying by alleviating the information asymmetry [26], thus lowering the trade barriers, breaking the barriers of local protectionism to commodity circulation, promoting the circulation of commodities among regions, and also contributing to the flow and aggregation of factors among regions and optimizing the allocation of factors. Third, the incentive policy brought about by the establishment of the National Big Data Comprehensive Experimental Zone. From the perspective of resource intensive, the establishment of the experimental zone helps the real economy to precipitate high-quality production factors and form the foundation of digital capability by gathering resources such as capital, information, and technology. The computing power resources contained in digital technology will promote the formation of intensive and efficient resource development patterns. Based on the above analysis, this paper puts forward the following research hypotheses:

**Hypothesis** **1** **(H1).**
*The establishment of the National Big Data Comprehensive Experimental Zone helps to improve the green total factor productivity.*


### 2.4. Big Data, Factor Market Distortions, and Green Economic Development

Big data can promote the development of a green economy by improving the allocation efficiency of production factors such as labor and capital. First, big data affects the development of a green economy mainly from economic effects by improving the efficiency of labor allocation. The reduced mobility of labor under the household registration system and land system, resulting in inter-regional wage differences and labor market segmentation leads to labor market distortions. The distortion in the labor market not only causes a mismatch in the structure of labor supply and demand but also makes enterprises more inclined to invest in cheap labor and reduce R&D investment, thus inhibiting the improvement of total factor productivity (Dou and Ji, 2018) [27]. The development of big data has broken through the fetters of the dual urban-rural household registration system on the free flow of labor to realize distributed employment across time and geographical distance. Under the new employment mode, the idle labor force and scattered labor force are fully utilized, the per capita labor output is improved, and the potential for economic growth is stimulated (Yang, 2017; Chen and Chang, 2022) [28,29]. Second, big data affects the development of a green economy from economic and environmental effects by improving the efficiency of capital allocation. In terms of economic effects, capital market distortion will make the gap between the return of capital factors and other factors increase, there are significant financing differences among enterprises with different ownership, and the increase in financing difficulty will reduce the incentive of product innovation of enterprises to a certain extent (Midrigan and Xu, 2014) [30]. As an important part of big data, digital inclusive finance relies on big data and cloud computing to collect, analyze, and make decisions on enterprise basic information big data in real-time, effectively reducing the cost of searching, trading, and matching, promoting the flow of factors in a larger space, ensuring the financing needs of small and medium-sized enterprises, alleviating capital distortion (Tang et al., 2019) [31], and improving the independent innovation ability of enterprises. Environmental effects are mainly reflected in overcapacity and environmental pollution. Big data alleviates the discrimination of enterprise ownership and enterprise scale in the capital market, which reduces the large number of redundant constructions caused by capital distortion and solves the problem of overcapacity (Bai, 2019; Zhang and Fan, 2022) [32,33]. Based on the above analysis, this paper puts forward the following research hypotheses:

**Hypothesis** **2** **(H2).**
*The establishment of the National Big Data Comprehensive Experimental Zone increases the green total factor productivity by improving the distortion of labor and capital factors.*


## 3. Research Design

### 3.1. Measurement Model Setting

In this paper, the establishment of a national big data comprehensive experimental area is taken as a quasi-natural experiment. On the policy time node, the policy node of Guizhou Province is set to 2015, and the policy nodes of other experimental areas are set to 2016. A total of 67 experimental areas are taken as experimental groups, and the rest of the cities are taken as control groups. The policy effect of the establishment of experimental areas on regional green total factor productivity is identified by DID with multiple time periods. The benchmark model is set as follows:(1)$$GTFP_{it}=\alpha_{0}+\alpha_{1}DID_{it}+\alpha_{2}X_{it}+\varphi_{i}+\mu_{t}+\varepsilon_{it}$$
where GTFP is urban green total factor productivity; $DID_{it}$ indicates whether the city *i* is in the state-level big data comprehensive experimental area in the *t* year virtual variable; The value of the year in which the National Big Data Comprehensive Experimental Zone is set up and the year after that is 1, otherwise, the value is 0; $\alpha_{1}$ depicts the influence of experimental area on green total factor productivity; $\varepsilon_{it}$ represents random disturbance term. In addition, this paper also controls the urban fixed effect $\varphi_{i}$ and time fixed effect $\mu_{t}$. Considering the possible bias of the estimation result caused by missing variables, this paper adds a series of control variables $X_{it}$, including Fiscal decentralization (*Fiscal*, expressed by the ratio of budgetary revenue to budgetary expenditure), actual utilization of foreign capital (*Fdi*, expressed by the ratio of actual foreign capital used in that year to regional GDP), Urbanization level (*Urban*, expressed by the logarithm of total population at the end of the year), industrial structure (*Ind*, expressed by the ratio of added value of secondary industry to regional GDP), and R&D investment (*Sic*, measured by the ratio of scientific expenditure to regional GDP).

### 3.2. Description of Green Total Factor Productivity Index

Green total factor productivity (*GTFP*) is different from total factor productivity (*TFP*). It comprehensively considers unexpected outputs such as pollution discharge and ecological environment factors, and evaluates the quality of economic development more comprehensively and reasonably, which is in line with the concept of contemporary green development. Green total factor productivity adds pollution emissions into the calculation framework of total factor productivity, and for a long time, the general data envelopment analysis (DEA) was used to incorporate resources and environmental factors into the production function analysis. However, this method can’t reflect the directionality of unexpected output and expected output, and the strict requirements of radial and angular further restrict the application scope of the directional distance function (Yue et al., 2018) [34]. To this end, Fukuyama and Weber (2008) [35] constructed a new directional distance function (SBM-DDF) based on slack variables based on DEA, which solves the problem of bias in the measurement of input-output variables brought about by the entry of pollutant variables into the production function. Considering that the SBM directional distance function can comprehensively measure the maximum and minimum frontier distance of expected output and unexpected output, the GML index can avoid no solution of linear programming while considering unexpected outputs. In order to enhance the rationality of the research conclusion, after comparing various measurement methods, the SBM-GML function is used to measure the green total factor productivity, with 2007 as the base period and logarithm. The specific measurement indicators are as follows:

(1) Factor inputs: including capital, labor, and energy inputs, of which capital inputs are estimated by the classical perpetual inventory method; labor input is expressed by the number of employees in urban units at the end of each year; energy input is expressed by urban electricity consumption, but considering the lack of data in some cities, Chen (2022) [36] is used to calculate the urban electricity consumption based on the calibrated night light data.

(2) Output: including expected output and unexpected output. The expected output is expressed by real GDP excluding the influence of price fluctuations. The unexpected output is the discharge of pollutants, including the discharge indicators of industrial wastewater, carbon dioxide, and soot. The threshold method is used to construct comprehensive pollutant discharge indicators.

### 3.3. Data Sources

The data in this paper are panel data of 284 cities from 2007 to 2020, and the relevant data come from the *China City Statistical Yearbook*, statistical bulletins of various cities, and statistical bureaus of various cities. The economic growth rate in 2007 was the highest since 1995, and the discharge of major pollutants showed a downward turning point for the first time. Therefore, 2007 was taken as the starting point of the study. At the same time, the samples are processed in the following aspects: First, the cities with data missing of more than 90% in the sample period from 2007 to 2020 are excluded. Second, for some missing sample data, we refer to the practices in Appendix 2 of *China Digital Economy Development White Paper (2021)* and use the mixed dynamic factor algorithm to supplement some missing data. Ultimately, a cumulative total of 3977 observations were obtained through data collection processes. The data processing aspect was executed utilizing Stata16 software.

## 4. Analysis of Benchmark Empirical Results

### 4.1. Benchmark Regression Analysis

#### 4.1.1. DID with Multiple Time Periods Results

The test results of the impact of the establishment of the National Big Data Comprehensive Experimental Zone on green total factor productivity are shown in Table 1.

The regression results presented in Columns (1) and (2) of Table 1 are estimated using a mixed least squares (POLS) approach based on the equation specified in Formula (1). Column (1) reveals that the coefficient of DID is significantly positive at a confidence level of 5%. Column (2) extends the analysis by adding control variables to the model in Column (1), resulting in an unchanged regression coefficient that remains significantly positive at a 5% confidence level. To mitigate the impact of omitted variables and increase computational efficiency, a multidimensional panel fixed effect model (reghdfe) is employed for estimation. The results presented in Columns (3) and (4) of Table 1, derived from the multidimensional fixed effect model, are in general agreement with those in Columns (1) and (2). The findings suggest that the establishment of the National Big Data Comprehensive Experimental Zone has a significant positive impact on green total factor productivity, proving Hypothesis 1.

#### 4.1.2. Balance Trend Test

The premise of the consistency of the DID model results is that the experimental group and the control group meet the parallel trend hypothesis, that is, before the implementation of the pilot policy, the trend in green total factor productivity in pilot cities and non-pilot cities should be parallel. Therefore, this paper refers to the time study method proposed by Jacobson et al. (1993) [37] to test the dynamic effect of the pilot policy in parallel trend, and constructs the model as follows:(2)$$GTFP_{it}=\beta_{0}+\sum_{t=-4}^{3}\beta_{t}treat_{i}\times post_{t}+\beta_{4}X_{it}+\varphi_{i}+\mu_{t}+\varepsilon_{it}$$
where $treat_{i}\times post_{t}$ is the interaction term obtained by multiplying the time dummy variable *Post* and the grouping dummy variable *Treat* before and after the establishment of the experimental area. The other variables are the same as those of the benchmark model, so we should pay attention to the coefficient $\beta_{t}$ in this model. If the cross-product coefficient before the establishment of the experimental area is not significant, it means that there is no significant difference between the experimental group and the control group before the establishment of the experimental area, and the two groups of samples have parallel trends, which meet the requirements of the model.

Figure 1 shows the estimated result of $\beta_{t}$ with a 95% confidence interval. Figure 1a shows OLS regression results, and Figure 1b shows high dimensional fixed effects regression results, which are basically consistent with each other. In this paper, it is found that the estimated coefficient of the interaction term $\beta_{t}$ in each period before the establishment of the pilot area is not significant, which indicates that there is no significant difference between cities before and after the establishment of the pilot area, and the research samples have passed the parallel trend test. In addition, after the establishment of the pilot area, the estimated coefficient $\beta_{t}$ increased significantly and gradually from the third year, indicating that the impact of the pilot policy on green total factor productivity increased gradually two years later.

### 4.2. PSM-DID Test

After the above balance trend test, the propensity score matching method was introduced to screen out the experimental group and the control group with a high matching degree to reduce the influence of non-random selection bias of DID with multiple time periods.

#### 4.2.1. Propensity Score Matching Kernel Density Results

A kernel density plot can be used to examine PSM quality. The more overlapping parts of the experimental group and control group in the kernel density plot, the better the matching effect. Figure 2 shows that after the PSM test, the nuclear density distribution of the experimental group and the control group almost completely overlap, indicating that the matching quality is good. This has laid a good data foundation for this paper to further explore the impact of the establishment of the National Big Data Comprehensive Experimental Zone on green total factor productivity by using the DID method.

#### 4.2.2. Propensity Score Matching Equilibrium Test

In order to make PSM results more reliable, the results should meet the “conditional independence assumption”, which requires that there is no significant difference between the experimental and control groups on the matched variables. The results in Table 2 show that the coefficients of the matched control variables are not significant, that is, the mean values of the matched control variables are evenly distributed, indicating that the original hypothesis of equal means of the matched variables after matching was accepted and the PSM was valid.

#### 4.2.3. PSM-DID Test Results

Table 3 reports the PSM-DID empirical results of the impact of the establishment of the National Big Data Comprehensive Experimental Zone on green total factor productivity. In this paper, PSM is used to match the samples of the experimental group and the control group, and the K-order (k = 2) nearest neighbor matching method is selected, as shown in column (1) of Table 3. The DID coefficients are significantly positive at a 5% confidence level, that is, the establishment of the experimental zone still significantly improves the regional green total factor productivity. The PSM-DID model test results are basically consistent with the benchmark regression results, which further verifies the promotion effect of the establishment of the National Big Data Comprehensive Experimental Zone on the green total factor productivity.

### 4.3. Robustness Test

(1) Instrumental variables method. In order to avoid endogeneity from interfering with the regression results, suitable instrumental variables should be selected for testing. Referring to the practice of Zhao et al. (2020) [38], this paper uses the multiplication term of the number of Internet users in the whole country that lags behind one period and the number of fixed telephones per 10,000 people in various prefecture-level cities in 1984 as the instrumental variable of the big data development to make a two-stage estimation. On the one hand, traditional communication technology is the foundation of the development of the Internet, and the previous communication infrastructure construction level of the city will affect the subsequent digital construction of the city, meeting the relevant conditions; At the same time, the impact of the use of traditional communication tools on urban development will gradually fall into recession with the decrease in the frequency of use, which meets the exogenous conditions. The regression results in column (1) and column (2) of Table 4 all passed the test of insufficient and weak identification of instrumental variables, which showed that the conclusion that the establishment of a national-level big data comprehensive experimental zone had a positive impact on regional green total factor productivity was still valid after considering endogenous problems.

(2) Eliminate the influence of the “Broadband China” strategy. In benchmark regression, does a net effect exist in the relationship between the establishment of the National Big Data Comprehensive Experimental Zone and regional green total factor productivity? Will the improvement in regional green total factor productivity be interfered with by other related policies? Through sorting out the policy documents issued in 2015 and 2016, it is found that from 2014 to 2016, three batches of cities received the pilot opportunity of “Broadband China”. This paper holds that the pilot of “Broadband China” will further improve the regional green total factor productivity, and the economic and green effects of the national big data comprehensive experimental zones may be underestimated or overestimated. To identify and evaluate this impact, this paper adds the corresponding policy dummy variable *Web* (for the pilot cities, it is set as 1 in the year of establishment and later, and 0 in other years) on the basis of the benchmark model. It can be seen from the regression results in column (3) of Table 4 that the policy of “Broadband China” and the policy of setting up a national-level big data comprehensive experimental zones have significantly and positively influenced the regional green total factor productivity, and the DID coefficient is relatively improved. This conclusion indicates that the promotion effect of regional green total factor productivity brought by the establishment of the National Big Data Comprehensive Experimental Zone may be underestimated, but its influence effect still exists.

(3) Policy exogeneity test. DID with multiple time periods adopted in this paper has certain preconditions, which require the externality of the national big data comprehensive pilot area policy to be ensured, that is, the research object has not been effectively expected before the implementation of the policy. In this paper, the method of Sun and Song (2022) [39] is used for reference, and the dummy variable *DID_before* of the year before the establishment of the experimental area is added to the benchmark regression model. It can be seen from the regression results in column (4) of Table 4 that the *did* coefficient is consistent with the above, but the *DID_before* estimation coefficient is not significant, which indicates that there is no expected effect.

(4) Replace the explanatory variables. The SBM-DDF approach was used to re-measure regional green total factor productivity. The robustness test results in column (5) of Table 4 show that the DID coefficient is significant at the 5% confidence level as evidenced by the results of the benchmark regression, which is generally consistent with the results of the benchmark regression.

(5) Change the sample matching method in the PSM-DID test. In order to make the estimation result more reliable, this paper changes the matching method to re-match and uses radius matching, Mahalanobis distance matching, kernel matching, and other methods to match the samples, and then carries out DID estimation. Table 4 shows that after the PSM matching method is changed, the DID estimation results are close to the benchmark regression results, and the results are robust.

### 4.4. Robustness Test

Given the different resource endowments of cities in China, there are differences in the impact of the establishment of national big data comprehensive experimental zones on regional green total factor productivity, and it has significant practical significance and policy value to distinguish and identify the urban characteristics that lead to the differences. In view of this, this paper examines whether different urban characteristics affect the impact of the establishment of experimental zones and green total factor productivity, starting from the human and financial resources required for urban development. Specifically, referring to the practices of Zhan and Li (2022) [40], the level of human capital in cities is measured using the share of regional tertiary school enrolment in the total year-end population, the ratio of the deposit balance of financial institutions to the regional GDP is used to measure the city’s financial development level, and the average wage of employees and regional GDP are used to measure the city’s economic activity level. According to the median of the above three indicators, this paper divides the cities into two groups: high level and low level, and tests them respectively.

Table 5 shows that for cities with low human capital and financial resources, the establishment of national big data comprehensive experimental zones has no significant impact on regional green total factor productivity; However, cities with higher human capital and financial resources have a significant positive impact, which indicates that human capital, financial development, and economic activities are all important resource elements to support the high-quality economic development of cities that set up national-level big data comprehensive experimental zones. A strong economic foundation and financial support are the guarantees for the R&D and application of new-generation technologies and the long-term and orderly advancement of intelligent and digital city construction. The above results confirm that the impact of the establishment of national big data comprehensive experimental zones on regional green total factor productivity is heterogeneous among cities with different characteristics.

## 5. Further Analysis

Previous studies have shown that the establishment of national-level big data comprehensive experimental zones contributes to the improvement of regional green total factor productivity. The above research conclusions provide sufficient empirical evidence for clarifying the high-quality development of a big data-enabled economy. However, the black box of the mechanism that the establishment of national-level big data comprehensive experimental zones affects regional green total factor productivity has not yet been opened. This section will be devoted to exploring the internal mechanism of setting up the pilot zone to improve the regional green total factor productivity. In order to describe the specific channel mechanism that affects the green total factor productivity in the experimental area, this paper sets up the following equation for identification and testing:(3)$$GTFP_{it}=\theta_{1}+\theta_{2}DID_{it}+\theta_{3}DID_{it}\times Abstaul_{it}+\theta_{4}Abstaul_{it}+\theta_{5}X_{it}+\varphi_{i}+\mu_{t}+\varepsilon_{it}$$
(4)$$GTFP_{it}=\theta_{1}+\theta_{2}DID_{it}+\theta_{3}DID_{it}\times Abstauk_{it}+\theta_{4}Abstauk_{it}+\theta_{5}X_{it}+\varphi_{i}+\mu_{t}+\varepsilon_{it}$$

In this paper, the labor distortion index (Abstaul) and the capital distortion index (Abstauk) are chosen as the moderating variables. On this basis, the paper focuses on the path of “mitigating factor allocation distortions”. The factor distortion index can be derived from economic equilibrium and is expressed as follows.
(5)$$Staul_{Li}=\frac{1}{1+\tau_{Li}},Stauk_{Ki}=\frac{1}{1+\tau_{Ki}}$$
where $L_{i}/L$ and $K_{i}/K$ are the ratio of actual labor to total labor and the ratio of actual capital to total capital in a city *i*, respectively. $S_{i}\beta_{Li}/\beta_{L}$ and $S_{i}\beta_{Ki}/\beta_{K}$ represent the ratio of labor to capital in a city *i* under effective allocation, respectively, and the relative distortion coefficient means the deviation degree between the factors actually used and those effectively allocated. Labor capital factor output elasticity $\beta_{Li}$ and $\beta_{Ki}$, are replaced by reference to the values of 0.44 and 0.56 measured by the Solow residual method in Zhao et al. (2006) [41].

Enabling technological innovation and upgrading industrial structure are prominent features of big data development. Columns (3) and (4) of Table 6 show that the establishment of a national big data comprehensive experimental zone has a significant positive impact on regional green total factor productivity, but the cross-product term between the capital distortion and the establishment of an experimental zone is significantly negative, which means that the capital factor distortion has a significant negative adjustment to the relationship between the establishment of an experimental zone and regional green total factor productivity. Columns (1) and (2) of Table 6 show the results of introducing the moderating variable labor distortion. Before the introduction of labor distortion variables to form a cross-term, the main effect of the experimental area on green total factor productivity is significantly positive. After the introduction of the cross-term of labor distortion index, the coefficient of the cross-term is significantly negative. This shows that digital development can alleviate the distortion in labor allocation and capital allocation, and then improve the green total factor productivity. According to the above theoretical analysis, the development of a digital economy characterized by big data can alleviate information asymmetry, reduce transaction costs, promote innovation and upgrade industrial structure, and then effectively reduce barriers to the flow of labor and capital factors, optimize the allocation of factor resources, and improve green total factor productivity.

## 6. Conclusions and Discussion

### 6.1. Conclusions

China’s economy has transformed from a high-speed growth stage to a high-quality development stage, and the digital economy has become a new driving force for high-quality economic development. Aiming at the green value of the digital economy, based on the distorted perspective of factor allocation, this paper uses the panel data of 299 prefecture-level cities in China from 2007 to 2020, and adopts the DID with multiple time periods model and PSM-DID model to empirically test the impact of the establishment of national-level comprehensive big data pilot zones on green total factor productivity and explore the relationship between big data and green economic development, and obtains the following conclusions:

(1) The establishment of the National Big Data Comprehensive Experimental Zone has promoted the regional green total factor productivity, and the empowerment of big data can indeed promote the development of the green economy. After adopting a variety of robustness tests, this conclusion still holds, which proves hypothesis 1. Digital development has a positive impact on green transformation, which is consistent with the research conclusions of Cheng and Qian (2021) [42] and Dai et al. (2022) [43]. The digital economy can not only improve the production efficiency of China’s industrial sector but also has the role of low-carbon environmental protection. The green total factor productivity of China’s industry has been significantly improved. At the same time, at the micro-enterprise level, digital empowerment can promote the green transformation of enterprises.

(2) The distortion of the factor market inhibits the development of a green economy, and the establishment of national big data comprehensive experimental zones can improve the green total factor productivity by optimizing the distortions in capital allocation and labor allocation, which proves Hypothesis 2. The distortion of the factor market has a negative impact on green total factor productivity, which is consistent with the research conclusions of Xie (2019) [44] and Bian (2021) [45]. The development of a digital economy promotes the free flow of factors and realizes the optimization of resource allocation through information spillover, technology spillover, and knowledge spillover, which is consistent with the research conclusion of Ren et al. (2022) [46].

(3) From the perspective of the variability in impact effects, the establishment of national big data comprehensive experimental zones has significant heterogeneity on green total factor production and the establishment of national big data comprehensive experimental zones in areas with high levels of human capital, financial development, and economic activity has a more significant effect on improving green total factor productivity. This shows that big data empowerment is mainly based on the scale effect and the technical effect to promote economic green transformation.

### 6.2. Discussion

#### 6.2.1. Theoretical Contributions

The research on the relationship between the digital economy characterized by big data and green economic development is currently the focus of attention at the national policy level, but there are few topics related to the green development of the digital economy. Relative to the existing studies, the innovations existing in this paper are mainly as follows:

(1) The green economic development model based on big data gives consideration to both production efficiency and green development and expands the research perspective of high-quality economic development. The problem of environmental pollution is the result of long-term extensive economic growth, and it may lack practical significance to study environmental pollution simply without economic growth. Therefore, compared with previous studies that only focused on the single-level factors of economic development or environmental pollution, this study will bring economic growth and environmental pollution into a complete framework for analysis, which is bound to have a more comprehensive understanding of the impact of big data on green economic development.

(2) In terms of factor market distortion and green economic development, this study summarizes the factors affecting green economic development such as energy conservation and emission reduction policies, fiscal decentralization system, and R&D and innovation subsidy policies as the resultant factors of factor market distortion, and systematically explores how big data can affect green economic development by alleviating factor market distortion. Meanwhile, considering the characteristic facts of different economic development rates in different regions, we further analyze the heterogeneous impact of big data empowerment on green total factor production from the perspectives of human capital differences, financial development differences, and economic activity differences, and attempt to measure the possible transmission paths of the impact of big data empowerment on green economic development in depth from the scale effect and technology effect.

#### 6.2.2. Policy Implications

The research conclusion of this paper provides empirical evidence for the establishment of national big data comprehensive experimental zones to promote high-quality economic development, and has the following policy implications:

(1) Comprehensive implementation of the big data development strategy. Given the promotion of regional green total factor productivity after the establishment of the National Big Data Comprehensive Experimental Zone. Governments should further increase investment in R&D and application of new generation technologies and accelerate the construction of big data comprehensive experimental zones. For the provinces that have been set up as an experimental zone, it is necessary to further improve the relevant laws and regulations, guide and consolidate the new impetus of digital economy development, and amplify the radiation-driving effect of the experimental zone; For provinces that have not set up pilot zones, it is necessary to deploy big data strategic planning in combination with their actual economic development and resource agglomeration capacity to promote the coordinated development of the regional digital economy with pilot zones as the engine. (2) Promote the establishment of big data experimental zones according to local conditions, and seek regional balanced development. The construction of an urban big data experimental area should follow a city’s own development conditions, accumulate solid human resources and economic foundation, create a favorable policy environment, and promote dynamic and differentiated digital city strategy for the purpose of regional coordinated development to match the digital economy construction with urban manpower, financial resources, and material resources, and provide strong support for effectively solving the unbalanced and uncoordinated problems of regional development. (3) Factor distortion refers to a situation wherein the marketization process of the factor market lags behind that of the product market. The control and regulation of pricing power and distribution power of factor market by local governments will not only affect the technological innovation of big data empowerment and the upgrading of industrial structure but also seriously hinder the high-quality development of the local economy in the long run. Therefore, governments at all levels should play the role of factor allocation in the digital economy, reduce the flow barriers of production factors, and open up favorable conditions for empowering the digital economy and improving regional green total factor productivity.

## Figures and Tables

**Figure 1 ijerph-20-04097-f001:**
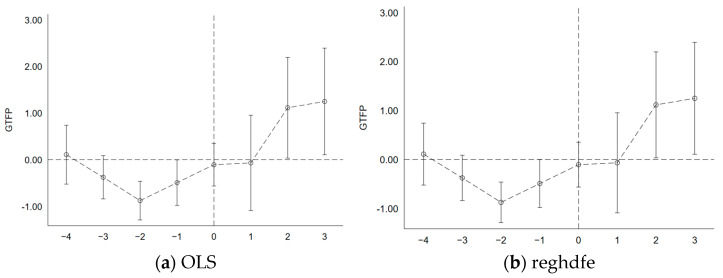
Balance trend test.

**Figure 2 ijerph-20-04097-f002:**
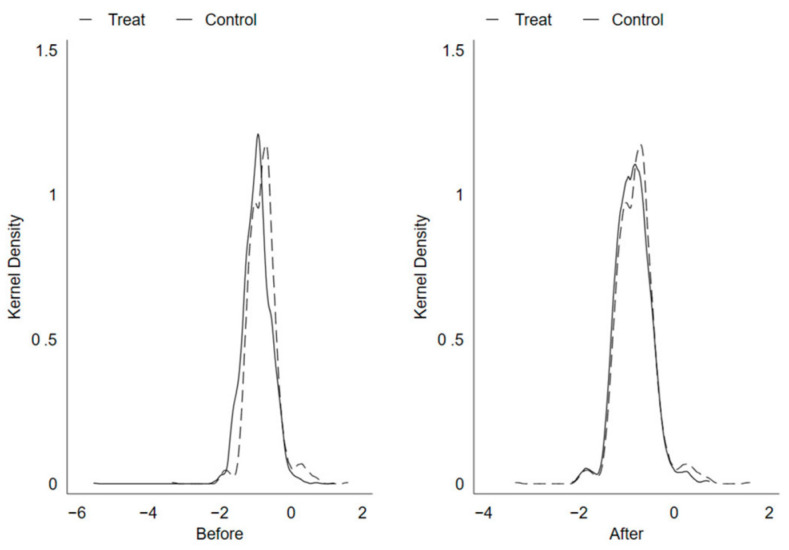
Kernel density function before and after tendency score matching.

**Table 1 ijerph-20-04097-t001:** Impact of Big Data Experimental Zone on Green Total Factor Productivity: DID with multiple time periods.

Variables	POLS		Reghdfe	
	(1)	(2)	(3)	(4)
*did*	0.086 **	0.077 **	0.086 **	0.077 **
	(2.185)	(1.988)	(2.274)	(2.071)
*Fiscal*		0.119		0.119
		(1.118)		(1.165)
*Fdi*		−1.520 *		−1.520 *
		(−1.779)		(−1.853)
*Urban*		0.049		0.049
		(0.411)		(0.428)
*Ind*		−0.002		−0.002
		(−1.408)		(−1.467)
*Sic*		0.089		0.089
		(0.126)		(0.131)
*Constant*	9.960 ***	9.807 ***	10.009 ***	9.786 ***
	(534.695)	(18.169)	(2965.345)	(14.390)
*Time fixed effect*	YES	YES	YES	YES
*Urban fixed effect*	YES	YES	YES	YES
*N*	3640	3539	3640	3537
*R* ^2^	0.198	0.203	0.198	0.203

Note: (1) ***, **, and * represent the significance level of 1%, 5%, and 10% respectively; (2) The T value adjusted by robust standard error (Cluster clustering to city level) is in brackets. The following are the same and will not be repeated here.

**Table 2 ijerph-20-04097-t002:** Results of balance tests for control variables before and after propensity score matching.

		Treat Group Mean	Control Group Mean	Differential	T-Value	*p*-Value
*Fiscal*	*Before*	0.498	0.466	14.4	3.79	0.000
	*After*	0.498	0.513	−6.9	−1.51	0.130
*Fdi*	*Before*	0.022	0.017	25.7	7.24	0.000
	*After*	0.022	0.021	1.7	0.38	0.706
*Urban*	*Before*	6.033	5.846	27.9	7.68	0.000
	*After*	6.033	6.049	−2.4	−0.58	0.561
*Ind*	*Before*	48.181	47.914	2.6	0.69	0.490
	*After*	48.181	48.249	−0.7	−0.15	0.878
*Sic*	*Before*	0.0311	0.032	−4.5	−1.21	0.225
	*After*	0.031	0.031	−1.1	−0.28	0.776

**Table 3 ijerph-20-04097-t003:** Estimation results of DID coefficients of different matching methods.

	(1)	(2)	(3)
	Radius Matching	Marginal Distance Matching	Kernel Density
*did*	0.076 **	0.077 **	0.077 **
	(2.066)	(2.071)	(2.157)
*Fiscal*	0.116	0.119	0.053
	(1.059)	(1.165)	(0.526)
*Fdi*	−1.559 *	−1.520 *	−1.502
	(−1.789)	(−1.853)	(−1.614)
*Urban*	0.054	0.049	0.161
	(0.397)	(0.428)	(1.441)
*Ind*	−0.002	−0.002	-0.001
	(−1.387)	(−1.467)	(−1.044)
*Sic*	0.007	0.089	−1.738 ***
	(0.005)	(0.131)	(−2.781)
*Constant*	9.760 ***	9.786 ***	9.182 ***
	(12.452)	(14.390)	(13.866)
*Time fixed effect*	YES	YES	YES
*Urban fixed effect*	YES	YES	YES
*N*	3535	3537	3533
*R* ^2^	0.203	0.203	0.207

Note: (1) ***, **, and * represent the significance level of 1%, 5%, and 10% respectively; (2) The T value adjusted by robust standard error (Cluster clustering to city level) is in brackets. The following are the same and will not be repeated here.

**Table 4 ijerph-20-04097-t004:** Robustness test results.

	(1)	(2)	(3)	(4)	(5)
	First Stage (DID)	Second Stage (GTFP)	Exclusion of Other Policy Factors	Policy Exogenous	Replace GTFP
*DID*		1.146 **	0.081 **	0.086 **	0.030 **
		(2.12)	(2.176)	2.23	(2.122)
*IV*	0.072 ***				
	(6.06)				
*Web*			0.062 **		
			(2.237)		
*DID_before*				0.078	
				(1.63)	
*Time fixed effect*	YES	YES	YES	YES	YES
*Urban fixed effect*	YES	YES	YES	YES	YES
*N*	3357	3357	3537	3539	3537
*R* ^2^	0.449	0.044	0.204	0.203	0.196

Note: (1) *** and ** represent the significance level of 1% and 5% respectively; (2) The T value adjusted by robust standard error (Cluster clustering to city level) is in brackets. The following are the same and will not be repeated here.

**Table 5 ijerph-20-04097-t005:** Heterogeneity test results.

	(1)	(2)	(3)	(4)	(5)	(6)
	High Level of Human Capital	Low Level of Human Capital	High Level of Financial Development	Low Level of Financial Development	High Level of Economic Activity	Low Level of Economic Activity
*did*	0.085 ***	0.073	0.130 *	0.055	0.109 **	0.035
	(2.834)	(1.077)	(1.852)	(1.560)	(2.586)	(0.615)
*Fiscal*	−0.066	0.277	0.169	0.041	−0.028	0.128
	(−0.628)	(1.653)	(0.782)	(0.355)	(−0.205)	(0.895)
*Fdi*	−0.444	−2.020 *	−2.504	−0.521	0.484	−2.861 **
	(−0.540)	(−1.949)	(−1.500)	(−0.863)	(0.610)	(−2.549)
*Urban*	0.392 *	0.060	0.034	0.091	0.010	0.002
	(1.717)	(0.523)	(0.206)	(0.634)	(0.068)	(0.010)
*Ind*	−0.003 *	0.002	−0.003	−0.002	−0.003 **	−0.003
	(−1.954)	(0.825)	(−1.176)	(−0.785)	(−2.197)	(−1.032)
*Sic*	0.699	−1.910 **	0.524	−1.932 ***	0.809	−3.147 ***
	(0.929)	(−2.035)	(0.732)	(−2.864)	(1.063)	(−3.434)
*Constant*	7.816 ***	9.552 ***	9.899 ***	9.593 ***	10.071***	10.253 ***
	(5.846)	(13.732)	(9.903)	(11.173)	(12.758)	(8.699)
*Time fixed effect*	YES	YES	YES	YES	YES	YES
*Urban fixed effect*	YES	YES	YES	YES	YES	YES
*N*	1711	1810	1712	1787	1672	1849
*R* ^2^	0.197	0.219	0.231	0.222	0.186	0.238

Note: (1) ***, **, and * represent the significance level of 1%, 5%, and 10% respectively; (2) The T value adjusted by robust standard error (Cluster clustering to city level) is in brackets. The following are the same and will not be repeated here.

**Table 6 ijerph-20-04097-t006:** Channeling mechanisms for the establishment of the pilot zone to affect green total factor productivity: mitigating distortions in the allocation of labor and capital factors.

	(1)	(2)	(3)	(4)
	Labor Mismatch	Labor Mismatch	Capital Mismatch	Capital Mismatch
*did*	0.031	0.028	0.047 *	0.056 **
	(1.528)	(1.426)	(1.859)	(2.220)
*did_staul*	−0.076 *	−0.070 *		
	(−1.890)	(−1.726)		
*abstaul*	0.049	0.072		
	(1.110)	(1.467)		
*did_stauk*			−0.163 **	−0.202 ***
			(−2.334)	(−2.873)
*abstauk*			−0.012	−0.008
			(−0.372)	(−0.258)
*Constant*	9.953 ***	9.838 ***	9.972 ***	9.731 ***
	(681.433)	(43.778)	(1457.815)	(41.856)
*Controls*	NO	NO	NO	NO
*Time fixed effect*	YES	YES	YES	YES
*Urban fixed effect*	YES	YES	YES	YES
*N*	3640	3537	3640	3537
*R* ^2^	0.104	0.112	0.104	0.113

Note: (1) ***, **, and * represent the significance level of 1%, 5%, and 10% respectively; (2) The T value adjusted by robust standard error (Cluster clustering to city level) is in brackets. The following are the same and will not be repeated here.

## Data Availability

Data are available in a publicly accessible repository. The data presented in this study are openly available in [FigShare] at [10.6084/m9.figshare.21804348].

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
