# Peer review of "The Role of Big Data in Promoting Green Development: Based on the Quasi-Natural Experiment of the Big Data Experimental Zone"

_ijerph, 2023, doi:10.3390/ijerph20054097_

Round 1

Reviewer 1 Report

What is the size of the data used in this study?  what are referred to as OLS and reghdfe in Table 1 and Figure 1?  They shall be introduced and explained.

Author Response

We would like to thank you for your careful reading, helpful comments, and constructive suggestions, which has significantly improved the presentation of our manuscript.

Point 1: What is the size of the data used in this study? what are referred to as OLS and reghdfe in Table 1 and Figure 1? They shall be introduced and explained.

Response 1: Thank you so much for your careful check. We have added more details. The data in this paper are for a panel of 284 prefecture-level cities from 2007 to 2020, resulting in a total of 3977 observations (in the first paragraph of subsection 3.3). OLS is estimated using mixed least squares and reghdfe is a multidimensional panel fixed effects model introduced to minimize the bias caused by omitted variables and to improve the speed of computation (in the second paragraph of subsection 4.1.1.).

Reviewer 2 Report

1.         The original abstract section asks the authors to add a description of the limitations of previous studies and the sources of the data.
2.         In addition, insights and implications should be added to the abstract section.
3.         In the literature review section, the review of previous studies is only a simple listing of others' findings, lacking an academic logic.      In addition, some important documents in the field of green development have been omitted.
e.g.,
Green innovation and organizational performance: The influence of big data and the moderating role of management commitment and HR practices.  Technological Forecasting and Social Change, 144, 483–498.  https://doi.org/10.1016/j.techfore.2017.12.016
How does the Belt and Road policy affect the level of green development?    A quasi-natural experimental study considering the CO2 emission intensity of construction enterprises.    Humanities and Social Sciences Communications, 9(1).    https://doi.org/10.1057/s41599-022-01292-4
The role of big data analytics in enabling green supply chain management: a literature review.   Journal of Data, Information and Management, 2(2), 75–83.   https://doi.org/10.1007/s42488-019-00020-z
Big Data Meet Green Challenges: Big Data Toward Green Applications.   IEEE Systems Journal, 10(3), 888–900.   https://doi.org/10.1109/jsyst.2016.2550530
etc.

4.         In this paper, green total factor productivity is chosen as the explanatory variable, where "3.2.          Description of green total factor productivity index" explains the method of this paper by referring to others' studies, however, both papers are old.
5.         Please explain why the authors only selected data for 2007-2020, and whether the authors can add the latest data for 2021 or 2022?
6.        The authors are invited to add a discussion section to discuss the commonalities or differences between the conclusions of this paper and those of previous studies, and to explain why.
7.         In the conclusion section, the author is suggested to add a paragraph to summarize all the conclusions.

To sum up, it is suggested that the authors carefully revise this manuscript according to the above comments. I sincerely look forward to receiving the revised version.

Author Response

We would like to thank you for your careful reading, helpful comments, and constructive suggestions, which has significantly improved the presentation of our manuscript.

Point 1: The original abstract section asks the authors to add a description of the limitations of previous studies and the sources of the data.

Response 1: Thank you for your advice. Following your suggestion, we have added a description of the limitations and data sources of previous research in the abstract.

Point 2: In addition, insights and implications should be added to the abstract section.

Response 2: Thanks for all your comments. In the revised abstract, we include insight and impact in the last sentence of the abstract.

Point 3: In the literature review section, the review of previous studies is only a simple listing of others' findings, lacking an academic logic. In addition, some important documents in the field of green development have been omitted.

Response 3: Thank you for the above suggestions. Following your suggestion, we improved the structure and organization of the manuscript. We revised Section 2, sorted out the literature according to the concept of "big data" and the academic logic of policy evolution-factor market distortion and green economy development-big data and green economy development, factor market distortion and green economy development, and then put forward relevant research hypotheses. In addition, we have joined the discussion of relevant documents in the field of green development in sections 2.2, 2.3 and 2.4.

Point 4: In this paper, green total factor productivity is chosen as the explanatory variable, where "3.2. Description of green total factor productivity index" explains the method of this paper by referring to others' studies, however, both papers are old.

Response 4: Thank you for your comment. In this study, the SBM-GML function is used to measure the green total factor productivity. This method is innovative on the basis of two classic papers, Chung et al. (1997), Fukuyama and Weber(2008), and GML index is used to consider the unexpected output and avoid the situation of linear programming without solution. The introduction of SBM-GML function will improve the comprehensiveness of the research on the development of green economy in our manuscript.

Point 5: Please explain why the authors only selected data for 2007-2020, and whether the authors can add the latest data for 2021 or 2022?

Response 5: Thank you for pointing out this problem in manuscript. The reason why we choose 2007 as the starting point of the research is that the economic growth rate in 2007 was as high as 11.4%, which was the highest growth rate since 1995. For the first time, the discharge of major pollutants showed a "turning point" of decline, and the overall environmental quality showed a trend of improvement. However, the situation of environmental protection Still grim. At this time, how to realize the green economic growth that "needs not only green water and lush mountains, but also golden mountains and silver mountains" has become an important issue of concern to all sectors of society. In addition, a large number of factor input and pollutant emission data of various cities in 2021 and 2022 are missing, and it is impossible to accurately calculate the regional green total factor productivity in 2021 and 2022. Bringing inaccurate data into the entire empirical test process will lead to lead to discrepancies in the empirical results.

Point 6: The authors are invited to add a discussion section to discuss the commonalities or differences between the conclusions of this paper and those of previous studies, and to explain why.

Response 6: Thank you for the above suggestion. Thank you for the advice. We changed the title of section VI to Conclusion and Discussion, and in the section 6.1. Conclusion we added the similarities between the conclusions of this paper and previous studies, and in the theoretical contributions section of 6.2.1. we emphasized the innovation of this study that is different from previous studies.

Point 7: In the conclusion section, the author is suggested to add a paragraph to summarize all the conclusions.

Response 7: We are appreciative of the reviewer’s suggestion. We have revised the structure and logic of section 6.1. In the first paragraph of section 6.1., we explain the process of empirical test of this paper, and the second to fourth paragraphs are further analysis of the research conclusion, comparing the conclusion of this paper with previous studies to improve the credibility and readability of the conclusion of this paper.

Reviewer 3 Report

The subject matter of the article and the research direction adopted are interesting and innovative. The research concept is clearly described by the authors in the introduction section. Similarly, the contribution of the conducted research to the current state of scientific knowledge has been convincingly described. The manuscript however needs thorough revision before resubmission and being suitable for publication at International Journal of Environmental Research and Public Health

The manuscript main issues are:

1.       I appreciate very much the design of the section: "literature review and research hypotheses" in the structure of the article. In theory, a section designed in this way makes it possible to clearly link the hypotheses adopted with the results of the work of other researchers. However, I find the literature review presented therein to be limited and modest. I believe that the literature review should be further developed.

2.       The paper lacks a discussion section, which should contain the most important findings of the research, a discussion of related research and a comparison between results obtained and initial hypotheses.

3.       The results of testing and verification of the adopted hypotheses should be precisely described by the authors with reference to the initial research assumptions.

4.       Tables 2-6 lack explanations of the * symbols used. This should be completed as for Table 1.

5.       All abbreviations used in the tables should be explained.

Author Response

We would like to thank you for your careful reading, helpful comments, and constructive suggestions, which has significantly improved the presentation of our manuscript.

Point 1: I appreciate very much the design of the section: "literature review and research hypotheses" in the structure of the article. In theory, a section designed in this way makes it possible to clearly link the hypotheses adopted with the results of the work of other researchers.  However, I find the literature review presented therein to be limited and modest.  I believe that the literature review should be further developed.

Response 1: We gratefully appreciate for your valuable suggestion. We revised Section 2, sorted out the literature according to the concept of "big data" and the academic logic of policy evolution-factor market distortion and green economy development-big data and green economy development, factor market distortion and green economy development, and then put forward relevant research hypotheses.

Point 2: The paper lacks a discussion section, which should contain the most important findings of the research, a discussion of related research and a comparison between results obtained and initial hypotheses.

Response 2: Considering the Reviewer’s suggestion, we have changed the title of the sixth section to conclusion and discussion. 6.1. Conclusion contains the most important findings in the study, the discussion of related studies and the comparison between the obtained results and the original hypothesis. 6.2. Discussion not only includes 6.2.1 Theoretical Contribution, which discusses the innovation points of this study that are different from previous studies, but also includes 6.2.2. Policy implications.

Point 3: The results of testing and verification of the adopted hypotheses should be precisely described by the authors with reference to the initial research assumptions.

Response 3: Thank you for your comments. In the revised manuscript, we reconstructed the manuscript, and focused on referring to the original research hypothesis in content, accurately describing the test and verification results of the adopted hypothesis, so as to highlight the logic of this manuscript.

Point 4: Tables 2-6 lack explanations of the * symbols used.  This should be completed as for Table 1.

Response 4: Thank you for pointing out this problem in manuscript. We have added the explanation of the * symbol used in Table 2-6.

Point 5: All abbreviations used in the tables should be explained.

Response 5: Thank you for pointing out this problem in manuscript. We have explained all abbreviations used in the table.

Round 2

Reviewer 2 Report

It is a pity that the authors did not revise the manuscript thoroughly as I suggested last time.

In particular, the authors added that the literature reviewed was not sufficiently relevant to the DID approach and was too distant from 2023.

Therefore, it is suggested that the authors reconsider the last opinion and carefully revise this manuscript again, especially paying attention to the effectiveness and criticality of the literature review.

Author Response

We would like to thank you for your careful reading, helpful comments, and constructive suggestions, which has significantly improved the presentation of our manuscript.

Point 1: It is a pity that the authors did not revise the manuscript thoroughly as I suggested last time. In particular, the authors added that the literature reviewed was not sufficiently relevant to the DID approach and was too distant from 2023. Therefore, it is suggested that the authors reconsider the last opinion and carefully revise this manuscript again, especially paying attention to the effectiveness and criticality of the literature review.

Response 1: Thank you for your advice. We have revised the text to address your concerns and hope that it is now clearer. Please see 2.2. Green economic development; 2.3. Big Data and Green Economic Development; 2.4. Big Data, Factor Market Distortions and Green Economic Development.

Reviewer 3 Report

The suggested revisions have been made. The paper is to accept.

Author Response

Thank you again for your positive comments and valuable suggestions to improve the quality of our manuscript.